# Anti-Inflammatory Effect of Korean Propolis on *Helicobacter pylori*-Infected Gastric Mucosal Injury Mice Model

**DOI:** 10.3390/nu14214644

**Published:** 2022-11-03

**Authors:** Moon-Young Song, Da-Young Lee, Young-Min Han, Eun-Hee Kim

**Affiliations:** College of Pharmacy and Institute of Pharmaceutical Sciences, CHA University, Seongnam 13488, Korea

**Keywords:** *Helicobacter pylori*, Korean propolis, anti-inflammation, NF-κB, gastric mucosal injury

## Abstract

Propolis, a natural resinous substance obtained from a variety of buds and plants, has been reported to possess various biological functions. Several recent studies have demonstrated the inhibitory effects of propolis on the growth of *Helicobacter pylori* (*H. pylori*) in vitro; however, current research efforts on Korean propolis (KP) remain insufficient especially in vivo. Our study aims to investigate the anti-inflammatory effect and molecular mechanism of KP on mouse gastric mucosa during *H. pylori* infection. We examined an in vivo *H. pylori*-induced gastric mucosal injury mice model. We found that KP inhibited the growth of *H. pylori* and attenuated the expression of *H. pylori* virulence factors such as cytotoxin-associated gene A, encoding urease A subunit, surface antigen gene and neutrophil-activating protein A. Moreover, KP reduced both gross lesions and pathological scores in *H. pylori*-challenged mice. In addition, KP markedly restrained the production of pro-inflammatory cytokines and nitric oxide levels compared with an untreated *H. pylori*-infected group. In particular, we found that KP repressed the phosphorylation of IκBα and NF-κB p65 subunit, and subsequently suppressed their downstream target genes. Taken together, these findings demonstrate the beneficial effects of KP on inflammation through the inhibition of NF-κB signaling as well as inhibition of *H. pylori* growth in a mouse model infected with *H. pylori*. This suggests the potential application of KP as a natural supplement for patient’s suffering from gastric mucosal injury caused by *H. pylori* infection.

## 1. Introduction

*Helicobacter pylori* (*H. pylori*) is a Gram-negative spiral-shaped bacillus that colonizes and spreads disease to the human stomach, affecting nearly half of the world’s population [1,2,3]. *H. pylori* infection leads to the pathogenesis of chronic gastritis, peptic ulcer and gastric adenocarcinoma [4,5]. Ever since the World Health Organization identified *H. pylori* as a high-risk carcinogen, considerable efforts have been devoted to therapeutic research aimed at combatting its harmful effects [6,7]. Notably, many studies have reported that the eradication and inhibition of *H. pylori* is particularly crucial for attenuating the pathogenesis of *H. pylori*-related diseases [8]. 

Nuclear factor-κB (NF-κB) is considered to be one of the major regulators of the inflammatory response, and can activate and amplify gastric disorders during *H. pylori* infection [9,10]. Interestingly, *H. pylori* mediates NF-κB activation through *H. pylori* virulence factors including cytotoxin-associated gene A (CagA) [11]. When infected with *H. pylori*, the inhibitors of NF-κB (IκBs) become phosphorylated [12]. In consequence, NF-κB is translocated from the cytosol to the nucleus, leading to the release of pro-inflammatory cytokines such as tumor necrosis factor-α (TNF-α), interleukin (IL)-1β and IL-8 [13,14]. This process ultimately results in strong exacerbation of gastric mucosal injury and inflammation [15,16,17]. Recently, several studies have provided evidence supporting the importance of natural compounds in the treatment of various diseases [18,19,20] and their effects on antibiotic resistance and inhibition of *H. pylori* growth [21,22,23]. 

Propolis is known as a natural remedy for *H. pylori* infection, and is produced and collected by honeybees [24,25]. Many studies have identified that propolis contains vital constituents of essential biologically active compounds such as flavonoids, caffeic acids, and phenolic esters [26]. Moreover, propolis has been shown to have anti-oxidant, anti-inflammation, anti-carcinogenic and anti-bacterial properties [27,28,29,30]. There are a variety of propolis-based mixtures for different pharmacological composites depending on the geographic region of extraction [28]. Therefore, the potency of propolis compounds also differs slightly depending on the country of origin. According to reports, Korean propolis (KP) contains a high total phenolic content and shows strong effects against various diseases [27,30,31]. In fact, our previous in vitro study demonstrated the anti-inflammation and anti-oxidant effects of KP against *H. pylori*-infected gastric damage [30]. Nevertheless, studies on the effects of KP against *H. pylori*-induced inflammation in animal models have not yet been published in the literature. In the present study, we demonstrate anti-inflammatory effects on gastric mucosal injury in experimental mice infected with *H. pylori* using KP samples obtained from 10 different regions in the Republic of Korea. 

## 2. Materials and Methods

### 2.1. Propolis Solution

The KP used in the present study was extracted, purified and collected from Korea Beekeeping Association (Republic of Korea) [30]. Ethanol extracts of KP samples were diluted with 0.5% carboxymethyl cellulose sodium salt (CMC) and adjusted to the required concentration (weight to volume), followed by storage in no-light conditions at 4 °C and warmed to room temperature before use.

### 2.2. Bacterial Culture

*H. pylori* Sydney Strain 1 (SS1) strain (Korea Collection for Type Culture, Daejeon, Republic of Korea) was used for the in vivo experiment. Bacteria were cultured in Tryptic soy agar with 5% sheep blood and incubated for 3–5 days under 10% CO_2_, 37 °C micro-aerobic conditions. *H. pylori* SS1 strain was collected and resuspended in cold phosphate buffered saline (PBS) in triplicate, and then used for the experiments.

### 2.3. Ethics

The mice experiments were reviewed and approved by the Institutional Animal Care and Use Committee of CHA University Animal Center (reference number: IACUC200110 and IACUC210159). The animals were handled in an accredited animal facility according to the guidelines and regulations for the use and care of animals of CHA University Animal Center (Seongnam, Republic of Korea).

### 2.4. Animal Experiment

C57BL/6 male mice (4-weeks old) were purchased from Orient Bio (Seoul, Republic of Korea). *H. pylori* SS1 strain was used for inoculation. Mice were acclimatized for 1 week and then orally administered with 0.2 mL of 5% sodium bicarbonate (as inhibitor of gastric acid) for 3 days and fasted for 12 h before *H. pylori* infection. *H. pylori* cultured in all groups except the normal group was adjusted to the number of bacteria of 5.0 × 10^9^ Colony-Forming Unit (CFU)/mL *H. pylori* suspension, four times at 2-day intervals. Normal group was orally administered an equivalent volume of PBS. All mice were handled in an experimental setting under a 12 h light/dark cycle and specific pathogen-free conditions. 

To confirm the maintenance of *H. pylori* infection, serum was isolated one-week later from venous blood extracted from the tails of all mice. *H. pylori* numbers were measured using the *H. pylori* IgG antibody ELISA kit (Cusabio Biotech Co., Houston, TX, USA), and mice with elevated *H. pylori* IgG levels were subsequently selected. Next, the mice were divided into three groups: Normal group; *H. pylori* group orally administered with 0.2 mL/kg PBS, 3-times weekly over a 4-week period, performed in triplicate; KP-treated group orally administered with 0.2 mL/kg of 200 mg/kg KP, 3-times weekly over a 4-week period.

### 2.5. H. pylori Antigen Test in Mouse Serum 

The *H. pylori* antigen was calculated using the *H. pylori* IgG antibody ELISA kit (Cusabio Biotech Co., Houston, TX, USA), according to the manufacturer’s instructions. Samples were diluted with a solution and incubated at room temperature for 30 min and then 100 µL was added to the *H. pylori* antigen. The results were detected after 30 min. All samples were read on 450 nm absorbance. 

### 2.6. Campylobacter-Like Organism (CLO) Test

The gastric mucosa tissue was determined using CLO test kit (Asan Pharmaceutical Co., Seoul, Republic of Korea). After incubation for 2 h at 37 °C, infection of *H. pylori* in mice produced a positive reaction, changing the color from yellow to purple. All samples were scored using the following rate: no color change: 0 points; slightly red color: 1 points; light purple: 2 points; dark purple: 3 points.

### 2.7. Histological Analysis

For histopathological assessment, stomach portions were fixed in 10% formalin, embedded in paraffin blocks, sectioned to 4 μm, stained with hematoxylin-eosin (H&E) and Periodic Acid Schiff (PAS), and then the histopathological scores were evaluated. Histopathological scores estimated overall gastritis in the entire corpus and antrum region of mice including inflammatory cell infiltration, submucosa edema, damage of the surface epithelium, and total pathologic score. The degree of gastritis was checked in the previous studies, and the samples were calculated in terms of scores [32]. A score of 0 points was given if no pathology was observed for each item, 0.5 points for mild, and 1 point for moderate. Subsequently, three different researchers measured all pathological index cases to increase objectivity. 

### 2.8. Quantitative Reverse Transcriptase Polymerase Chain Reaction (qRT-PCR) Analysis

The qRT-PCR was performed on a ViiATM 7 Real-time PCR system (Applied Biosystems, Waltham, MA, USA) using Luna universal qPCR master mix (New England Biolabs, Beverly, MA, USA). The relative quantities of genes were calculated from triplicate samples after normalization to 18S ribosomal RNA (18S rRNA, as internal control). All oligonucleotide primers were purchased from Macrogen (Seoul, Republic of Korea) and are listed in Table 1 below.

### 2.9. Western Blot Analysis

The stomach tissues were homogenized with ice-cold cell lysis buffer (Cell Signaling, Danvers, MA, USA) plus phosphatase and protease inhibitors (Roche Applied Science, Mannheim, Germany), and then centrifuged. Western blot analysis was performed as previously described [33]. Proteins were visualized using an enhanced chemiluminescence system (Thermo Fisher Scientific, Waltham, MA, USA). The primary antibodies to detect pp65, p65, pp50, p50, p-IκBα, and IκBα used in this study were purchased from Cell Signaling Technology (Danvers, MA, USA). Antibodies for β-actin and CagA were purchased from Santa Cruz Biotechnology (Dallas, TX, USA). Antibodies for iNOS were purchased from BD Biosciences (Franklin Lakes, NJ, USA). 

### 2.10. Cytokine Measurement via Enzyme-Linked Iimmunoassay (ELISA)

The serum levels of IL-8, IL-1β and TNF-α were measured with an ELISA assay kit (R&D Systems, Minneapolis, MN, USA) following the manufacturer’s instruction. 

### 2.11. Measurement of Nitric Oxide (NO) Production

The mouse serum was quantified using a NO detection kit (iNtRON Biotechnology, Seongnam, Republic of Korea) following the manufacturer’s instruction. 

### 2.12. Statistical Analysis 

Results were expressed as the mean ± standard deviation (SD). Statistical analysis of the data was performed using GraphPad (GraphPad Software, San Diego, CA, USA). All other experiments were performed on individual mice. The statistical significance was analyzed by one-way analysis of variance (ANOVA). Statistical significance was accepted at *p* < 0.05.

## 3. Results 

### 3.1. Gastric Mucosal Therapeutic Effect of KP in H. pylori-Infected Mice 

To investigate the anti-*H. pylori* effect of KP, we established *H. pylori*-infected gastric mucosal injury mice model. KP (200 mg/kg) was orally administered three times a week over a four-week period after inoculated with an *H. pylori* (5.0 × 10^9^ CFU/mL) suspension four times in two-day intervals (Figure 1A). After examination, first, we performed a test to check *H. pylori* IgG levels in mice. Upregulation of IgG levels is a classic indicator of *H. pylori* infection and was found to be significantly higher than in non-infection groups [34]. As expected, *H. pylori*-infected group showed a high IgG level, whereas it was significantly inhibited in the KP-treated group compared with the *H. pylori*-infected group (Figure 1B). We also checked for *H. pylori* infection using the CLO test, a sensitive and rapid method of detection for *H. pylori* [35]. The CLO test was conducted on the extracted gastric mucosal to examine the cure rate. As shown in Figure 1C, the *H. pylori*-infected group produced a positive reaction (purple color), but the KP-administrated group displayed a negative reaction (yellow color). These data indicate that the KP-treated group significantly inhibited the growth of *H. pylori* compared with the *H. pylori*-infected group. Next, we investigated gross legions present in the gastric mucosal layers in each group. The gastric mucosa of the *H. pylori*-infected group were thicker than the normal group, indicating infiltration of inflammatory cells in the sub-mucosal and mucosal layers [36]. As shown in Figure 1D, inflammatory cells were significantly observed in the gastric sub-mucosa and mucosa of *H. pylori*-infected mice compared with normal mice, whereas these inflammatory cell infiltrations were markedly decreased in the KP-administrated mice (*p* < 0.05). Overall, the gross legions showed that *H. pylori*-infected mice rapidly developed edema, slight bleeding, and considerable destruction of gastric mucosal layer, but the KP-treated mice were protected from gastric mucosal damage (Figure 1D). The severity of *H. pylori*-infected gastric mucosal injury was determined according to the degree of surface epithelium damage, total pathological score, inflammatory cell infiltration, and sub-mucosal edema (Figure 1E). The degree of inflammatory histopathological scores was measured using the previous strategy [32]. As demonstrated by the H&E staining technique, the *H. pylori*-infected mice group showed significant damage to the gastric mucosal surface epithelium, whereas the KP-treatment group showed dramatic preservation of the gastric mucosal layer (Figure 1E). Next, we validated whether KP protects the gastric mucosal layer against *H. pylori* infection via PAS staining in gastric tissues. Gastric surface mucous cells are mucous-producing cells that protect against the corrosive properties of stomach acid. In particular, *H. pylori* infection caused damage to the mucosal cells and accelerated gastric mucosal injury [37]. As shown in Figure 1F, the *H. pylori* group showed significant loss of mucous cells in gastric mucosal layer tissue. However, the KP-treatment group showed abundant presence and restoration of mucous within the cells, and gave a positive reaction of PAS staining results as indicated by purple coloring in the gastric mucosal layer (Figure 1F). These results provide supporting evidence for the protective effects of KP on mucous cells and mucosal layer from *H. pylori*-infection. 

### 3.2. KP Attenuates H. pylori-Related Virulence Factors in H. pylori-Infected Mice

A previous study by Han et al. revealed the effect of KP against eradication of *H. pylori* using paper disc diffusion method [31]. Thus, we hypothesized the possibility of KP suppressing the expression of virulence factors in *H. pylori.* The virulence factors of *H. pylori* are associated with pathogenic responses, colony formation, immune system disruption and disease development [38]. To evaluate the mitigating effects of KP against the virulence factors of *H. pylori*, we conducted qPCR test for *H. pylori*-related genes and virulence factors. As shown in Figure 2A, the *H. pylori*-infected group demonstrated significantly increased expression levels of *H. pylori* virulence factors such as 16s rRNA, Sydney strain 1 (SS1), encoding urease A subunit (ureA), Surface antigen gene (SSA) and neutrophil-activating protein A (napA). In comparison, the KP-treatment group demonstrated extreme reduction and attenuation in mRNA expression levels of virulence factors. These results demonstrated that the KP-treatment group repressed gastric mucosal injury by inhibiting *H. pylori* growth and virulence factors of *H. pylori*. Next, we also confirmed the protein levels of CagA, pathogenicity factors of the bacterial pathogen *H. pylori* using gastric tissue extracts. As a result, the *H. pylori*-infected group showed increased CagA protein expression, whereas the KP-treatment group showed strongly decreased the protein levels (Figure 2B). Therefore, we considered that KP might regulate virulence factors concerning *H. pylori* infection in gastric tissue extracts. 

### 3.3. KP Restrains the Pro-Inflammatory Response and NO Production in H. pylori-Infected Mice 

To demonstrate the effects of KP on pro-inflammatory responses in *H. pylori*-infected gastric mucosal injury mice, we examined the production levels of pro-inflammatory cytokines including IL-8, TNF-α, and IL-1β via ELISA and qPCR analysis. As shown in Figure 3A, the secretory serum levels of IL-8, TNF-α and IL-1β were significantly augmented after *H. pylori* infection, while the KP-treatment group showed an inhibitory effect on pro-inflammatory cytokines. Furthermore, the mRNA levels of IL-8, TNF-α and IL-1β were up-regulated in the *H. pylori*-infected group; however, mRNA expression was down-regulated in the KP-treatment group (Figure 3B). Next, we investigated whether the anti-inflammatory effect of KP on *H. pylori*-infected gastric mucosal injury is caused by inhibiting the production of NO. We performed the protein expression of inducible nitric oxide synthase (iNOS) via Western blot analysis and secretory serum levels of NO production. As a result, expression levels of iNOS and NO were markedly increased in the *H. pylori*-infected group, whereas the KP-treatment group showed significantly decreased levels of iNOS and NO production despite active infection with *H. pylori* (Figure 3C,D). These data suggest that the anti-inflammatory effect of KP occurs by suppressing IL-8, TNF-α, IL-1β and NO production in *H. pylori*-infected gastric mucosal injury mice model. 

### 3.4. KP Regulates the NF-κB Signaling Pathway in H. pylori-Infected Mice

iNOS is mediated by NF-κB activation and correlated with pro-inflammatory responses [39,40,41]. Therefore, we hypothesized that the anti-inflammatory effect of KP in *H. pylori*-infected gastric mucosal injury may be regulated by NF-κB signaling. As shown in Figure 4A, the *H. pylori*-infected group showed increased protein expression of IκBα phosphorylation, but this level was significantly restrained in the KP-treatment group (Figure 4A). Moreover, phosphorylation of p65 was significantly promoted in the *H. pylori*-infected group, which was dramatically abrogated by treatment with KP (Figure 4B). In contrast, there was no change in the p50 phosphorylated by *H. pylori* administration or treatment with propolis (Figure 4B). Additionally, we validated the c-myc, A20 and A1a mRNA levels of NF-κB target genes, which are correlated with pro-inflammatory responses [42,43,44,45]. As expected, the *H. pylori*-infected group showed increased expression of these target genes; however, these were remarkably reduced in the KP-treatment group (Figure 4C).

## 4. Discussion

Our study results demonstrated the anti-inflammatory and anti-bacterial effects of KP in an *H. pylori*-infected mice model. Traditionally, *H. pylori* eradication therapy is commonly treated with proton-pump inhibitors and antibiotics [46,47]. However, antibiotic resistance presents a significant obstacle in the successful eradication of *H. pylori* [48,49]. Hence, in this study, we investigated an alternative therapeutic approach using a natural compound for the treatment and eradication of *H. pylori* infection. 

Propolis has anti-inflammatory properties in the treatment of various diseases. Croatian propolis was reportedly used to treat obese patients with hyperlipidemic disorders, following prior success in mice studies [50]. In addition, Brazilian green propolis was suggested as a good candidate to suppress acute inflammation against hepatocellular necrosis in rats [51] and also indicated for use as a target in asthma immunotherapy [52]. In addition, Kai Wang demonstrated the protective effect of Chinese propolis against dextran sulfate sodium-induced acute colitis in rats [53]. Interestingly, several studies have shown that a variety of propolis such as Brazilian [54], Italian [55], Turkish [56], Bulgarian [57], Chilian [58] and Korean [31] inhibited the growth of *H. pylori* in vitro. Notably, our previous study has been clarified, in that KP attenuates inflammation against *H. pylori*-infected gastric epithelial cells via the NF-κB signaling pathway in vitro. However, the effect of KP on the reduction of inflammation against *H. pylori* on in vivo mice models has not been reported yet. Therefore, the main objective of our study was to elucidate the ability of KP to reduce *H. pylori*-induced inflammation through inhibition of NF-kB in vivo. 

*H. pylori* infection is characterized by accelerated inflammation and damage to the gastric mucosa via the elaboration of virulence factors such as CagA, ureA, SSA and napA [59,60]. These virulence factors contribute to the flagella, chemotactic systems, ureases, and adhesions upon *H. pylori*, thereby affecting the inflammatory response and immune system through a variety of pathways [61]. For example, VacA is associated with immunosuppression of T cell activation and mediates NF-κB activity within targeted T cells [62,63]. Peptidoglycan enters host epithelial cells via T4SS and activates NF-κB signals for the acceleration of pro-inflammatory cytokines [64,65]. In particular, the *H. pylori* virulence factor CagA has been extensively studied for its inflammation responses. Several studies have noted that CagA conducts an important role in the acceleration of pro-inflammatory cytokines through the NF-κB pathway [15,66]. Indeed, pro-inflammatory cytokines such as TNF-α, IL-1β and IL-8 have been found to be triggered during *H. pylori*-infection and utilizing various mechanisms for the induction of inflammation [67]. 

The NO produced by L-arginine in response to *H. pylori* infection may accelerate the inflammatory processes in gastric mucosal damage [68,69]. NO is divided into a family of three distinct NOS isoforms including nNOS, eNOS and iNOS. It is well known that its two isoforms, nNOS and eNOS, are both calcium-dependent. nNOS is constitutively expressed in neurons of the nervous system and eNOS is mainly expressed in vascular endothelial cells [70]. The third calcium-independent isoform, iNOS, is expressed in response to inflammatory cytokines and bacterial virulence factors inducing the release of NO [40,71]. Over-production of NO triggers the pathogenesis of a variety of diseases including gastritis, and upregulation of iNOS levels has been observed in gastritis patients with *H. pylori* infection [72,73]. Interestingly, Nam et al. showed that iNOS levels were significantly diminished after inhibition and eradication of *H. pylori* [74]. Additionally, NO production activates NF-κB signaling, thereby accelerating the pathological conditions and inflammatory responses, implicated in gastric mucosal damage against *H. pylori* infection [30,41]. Here, we show that KP inhibited NF-κB signaling and significantly reduced the expression of iNOS and production of NO in *H. pylori*-infected mice, which may contribute to the treatment of gastric mucosal injury. 

The transcription factor NF-κB is one of the major mediators of the inflammatory response against *H. pylori* infection [75]. *H. pylori* activates NF-κB through the canonical pathway, namely IKK complex phosphorylation/degradation of the IκB inhibitor, leading to the nuclear translocation of NF-κB, and thereby resulting in the acceleration of its target genes [15,68,76]. In this study, we showed that KP reduced the phosphorylation of IκBα and p65, thereby repressing the translocation of p65 from the cytosol to the nucleus in *H. pylori*-infected gastric mucosal injury mice. These findings suggest that KP attenuated the pro-inflammatory target genes through inhibition of NF-κB in *H. pylori*-infected mice. As a result, KP dramatically abrogated *H. pylori* virulence factors as well as host inflammatory responses during *H. pylori* infection (Figure 5). 

Lastly, we investigated a human equivalent dose for the effect of KP using the recommended body surface area normalization method [77]. KP demonstrated a therapeutic effect against *H. pylori*-infected gastric mucosal injury at a dose of 200 mg/kg. Based on our studies, a daily KP dosage ranging from 900 to 1000 mg is required for a 60 kg subject. However, in order to clarify the exact concentration of human equivalent dose for the effect of KP on *H. pylori* infection, further studies on the minimum inhibitory concentration in vivo are needed. 

## 5. Conclusions 

In conclusion, we verified the anti-bacterial and anti-inflammatory effects of KP against *H. pylori* infection. KP attenuated *H. pylori* virulence factors and the production of pro-inflammatory cytokines in *H. pylori*-infected mice, thereby decreasing damage to the gastric mucosa. We provided a clarified molecular mechanism whereby NF-κB becomes inhibited following KP administration during active *H. pylori* infection. Therefore, our data suggest that KP may have potential therapeutic application against *H. pylori* infection in a gastric mucosal injury mice model.

## Figures and Tables

**Figure 1 nutrients-14-04644-f001:**
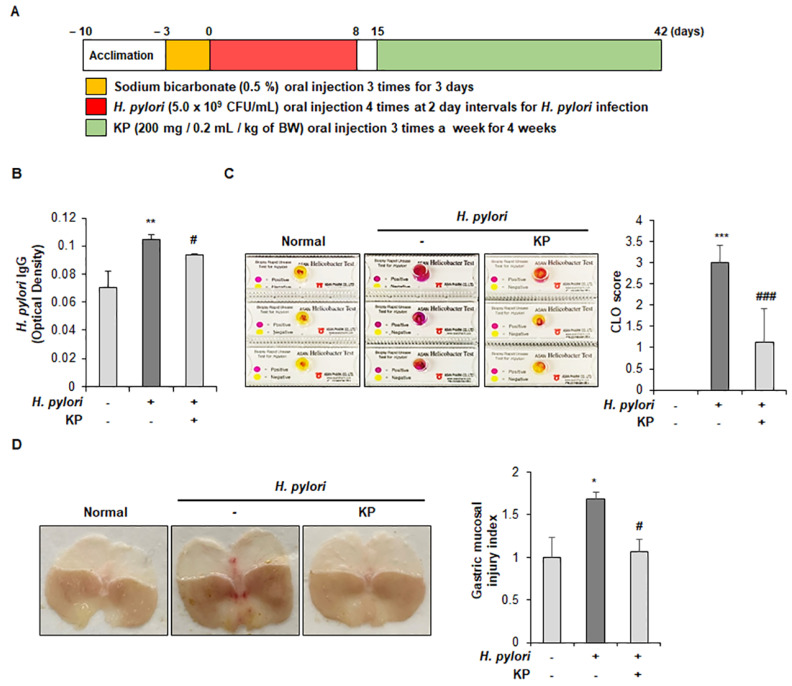
KP decreases the severity of *H. pylori*−infected gastric mucosal injury in mice. (**A**) Diagram shows the experimental design of *H. pylori*-infected gastric mucosal injury mice model. (**B**) Result of *H. pylori* IgG levels in mice using serum obtained from each group (*n* = 8). (**C**) Results of the CLO test in mice using gastric mucosal tissue from each group (*n* = 8, positive reaction shown by change in color from yellow to purple). (**D**) Representative images of the stomach and gastric mucosal injury index (*n* = 8). (**E**) Paraffin sections of stomach were stained with hematoxylin and eosin (H&E, magnification, ×100). Damage to the surface epithelium, inflammatory cell infiltration, submucosal edema and total pathological score in each group (*n* = 8) were quantified from H&E−stained sections. (**F**) Detection of mucous production was performed by PAS staining in each group (*n* = 3). Representative images of gastric sections were stained with PAS (magnification, ×100). Detection of mucous production were immunostained with PAS in mice gastric mucosal (positive result depicted by purple color), and one of three representative experiments is presented (×100). Graph shows the number of PAS-positive cells from at least 10 fields (bottom). Data are the mean ± standard deviation. Statistical significance was analyzed by analysis of variance. * *p* < 0.05, ** *p* < 0.01, and *** *p* < 0.001 vs. normal group; # *p* < 0.05, ## *p* < 0.01 and ### *p* < 0.001 vs. *H. pylori* group, significantly different compared with control.

**Figure 2 nutrients-14-04644-f002:**
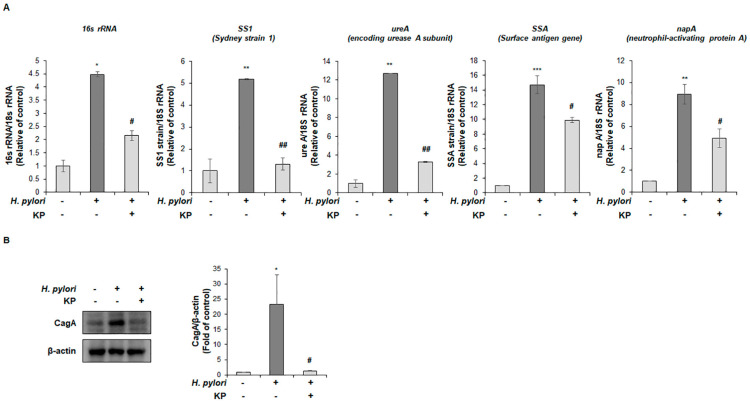
KP attenuates *H. pylori* virulence factors in *H. pylori*-infected gastric mucosal injury mice model. (**A**) mRNA expression levels in gastric mucosal tissues were examined via qRT-PCR. As an internal control, 18S rRNA was used for the expression of *H. pylori*-related genes. (**B**) The protein expression of CagA in gastric mucosal tissues was determined by Western blotting. β-actin was used as an internal control. Data are the mean ± standard deviation (*n* = 4). Statistical significance was analyzed by analysis of variance. * *p* < 0.05 and ** *p* < 0.01, and *** *p* < 0.001 vs. normal group; # *p* < 0.05 and ## *p* < 0.01 vs. *H. pylori* group, significantly different compared with control.

**Figure 3 nutrients-14-04644-f003:**
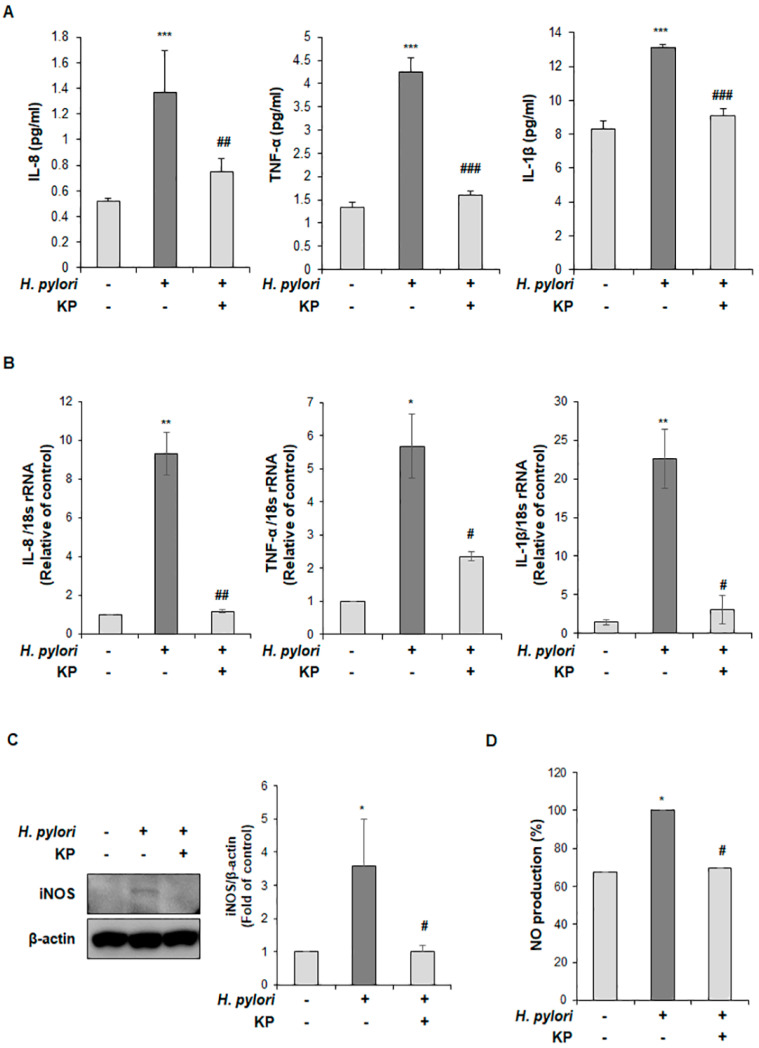
KP inhibits pro-inflammatory response and NO production in *H. pylori*-infected gastric mucosal injury mice model. (**A**) IL-8, TNF-α, and IL-1β production levels of pro-inflammatory cytokines were measured in serum via ELISA. (**B**) The mRNA expressions of IL-8, TNF-α, and IL-1β in gastric mucosal tissues were examined via qRT-PCR. The 18S rRNA was used as an internal control. (**C**) The protein expression of iNOS in gastric mucosal tissues was determined by Western blotting. β-actin was used as an internal control. (**D**) The secretory serum levels were measured for NO production. Data are the mean ± standard deviation (*n* = 4). Statistical significance was analyzed by analysis of variance. * *p* < 0.05, ** *p* < 0.01, and *** *p* < 0.001 vs. normal group; # *p* < 0.05, ## *p* < 0.01 and ### *p* < 0.001 vs. *H. pylori* group, significantly different compared with control.

**Figure 4 nutrients-14-04644-f004:**
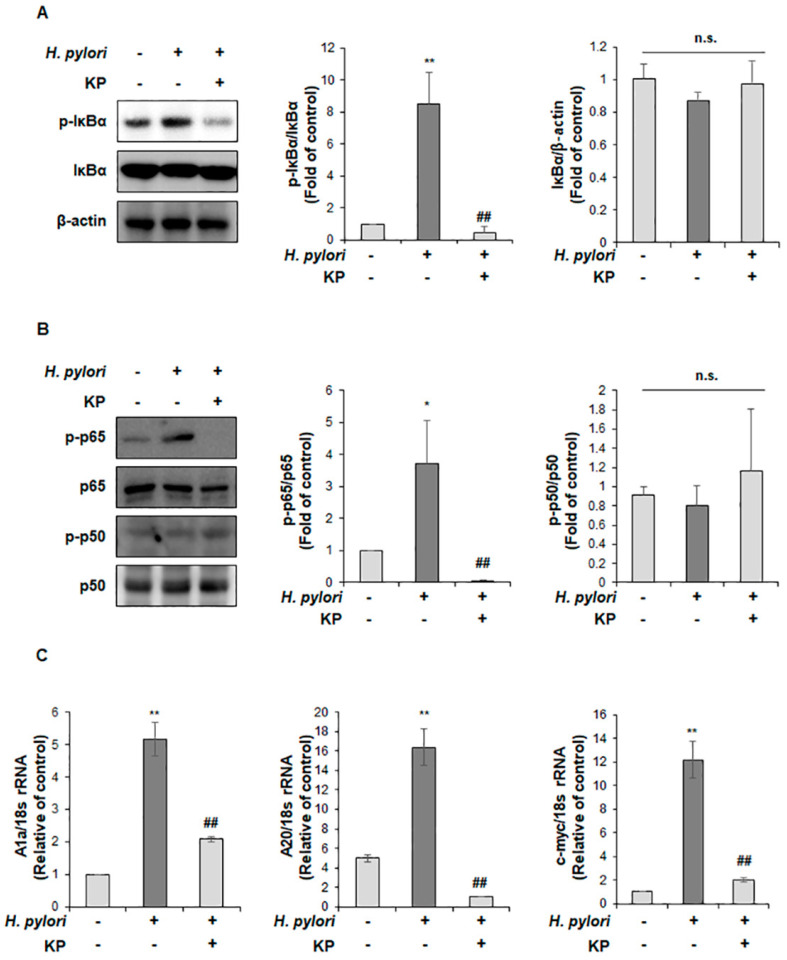
KP inhibits NF-κB signaling in *H. pylori*-infected gastric mucosal injury mice model. (**A**) Expressions of p-IκBα and IκBα in gastric mucosal tissues were analyzed by Western blot analysis. β-actin was used as an internal control. (**B**) The expression of protein levels of pp65, p65, pp50 and p50 were conducted by Western blot analysis. (**C**) The mRNA levels of A1a, A20 and c-myc were determined by qRT-PCR. The 18S rRNA was used as an internal control for the mRNA levels. Data are the mean ± standard deviation (*n* = 4). Statistical significance was analyzed by analysis of variance. * *p* < 0.05 and ** *p* < 0.01 vs. normal group; ## *p* < 0.01 vs. *H. pylori* group, significantly different compared with control.

**Figure 5 nutrients-14-04644-f005:**
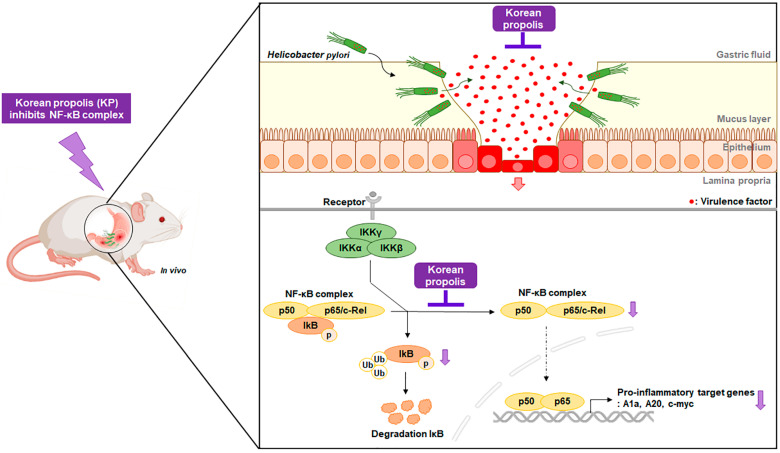
Schematic representation depicting KP inhibition of NF-κB-mediated inflammation in *H. pylori*-infected gastric mucosal injury in mice. “Black color ↓” indicates induction; “Purple color↓” indicates reduction; “├” indicates inhibition.

**Table 1 nutrients-14-04644-t001:** Mouse Primer sequences used for qRT-PCR.

Identification	Gene	Primer Sequence (5′ to 3′)	
Specific for *H. pylori*	*16S rRNA*	Forward	CTC ATT GCG AAG GCG ACC T
		Reverse	TCT AAT CCT GTT TGC TCC CCA
	*SS1*	Forward	CTT AAC CAT AGA ACT GCA TTT GAA ACT AC
		Reverse	GGT CGC CTT CGC AAT GAG TA
	*ureA*	Forward	AGG AAA CAT CGC TTC AAT ACC
		Reverse	AGG AAA CAT CGC TTC AAT ACC
	*SSA*	Forward	TGG CGT GTC TAT TGA CAG CGA GC
		Reverse	CCT GCT GGG CAT ACT TCA CCA TG
	*napA*	Forward	TCC TTT CAG CGA GAT CGT CA
		Reverse	GAA TGT GAA AGG CAC CGA TT
Specific for inflammation	*IL-8*	Forward	TCC TTG TTC CAC TGT GCC TTG
		Reverse	TGC TTC CAC ATG TCC TCA CAA
	*TNF-α*	Forward	TCA GAG GGC CTG TAC CTC AT
		Reverse	GGA AGA CCC CTC CCA GAT AG
	*IL-1β*	Forward	TTA AAG CCC GCC TGA CAG A
		Reverse	GCG AAT GAC AGA GGG TTT CTT
	*A1a*	Forward	TCC ACA AGA GCA GAT TGC CCT G
		Reverse	GCC AGC CAG ATT TGG GTT CAA AC
	*A20*	Forward	AGC AAG TGC AGG AAA GCT GGC T
		Reverse	GCT TTC GCA GAG GCA GTA ACA G
	*c-myc*	Forward	GCT GTT TGA AGG CTG GAT TTC
		Reverse	GAT GAA ATA GGG CTG TAC GGA G
Internal control	*18s rRNA*	Forward	GCA ATT ATT CCC CAT GAA CG
		Reverse	GGC CTC ACT AAA CCA TCC AA

qRT-PCR, quantitative reverse transcription PCR; *16S rRNA*, 16S ribosomal RNA; *SS1*, Sydney strain 1; *ureA,* encoding urease A subunit; *SSA*, Surface antigen gene; *napA*, neutrophil-activating protein A; *IL-8*, Interleukin-8; *TNF-α*, Tumor necrosis factor-α; *IL-1β*, Interleukin-1β; *18S rRNA*, 18S ribosomal RNA.

## Data Availability

The data presented in this study are available on request from the corresponding author.

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
