# Peer review of "Anti-Inflammatory Effect of Korean Propolis on Helicobacter pylori-Infected Gastric Mucosal Injury Mice Model"

_nutrients, 2022, doi:10.3390/nu14214644_

Round 1
Reviewer 1 Report
This manuscript describes the potential of using Korean propolis (KP) as an alternative therapeutic approach for the treatment of H. pylori infection. The authors aim to show that KP is able to inhibit the inflammation via suppression of NF-κB signaling on H. pylori-infected mice model. While some findings seem to be interesting, some puzzling features need to be further addressed.
1. As indicated in Fig.1A, KP was administrated after the mice was infected with H. pylori. Why the author stated that “these findings indicated that KP has “preventive effects” for inflammation via suppression of NF-κB signaling on H. pylori-infected mice model.” in the abstract? I do not agree that there is a prevented effect according to the design of your experiments.
2. Need to provide the full name for “CLO test”
3. The legend of Fig 1C is written as “Results of the CLO test with gastric mucosal following each group”. Since the resolution of Fig 1C is very poor, one cannot see the result at all even using zoon in.
4. In Figure 2, authors show that treatment of KP reduces the expression of H. pylori virulence factors in H. pylori-infected gastric mucosal injury mice model. Is the reduction due to the inhibition of the expression of these virulence factors in H. pylori or the eradication of H. pylori in the mice? This question needs to be addressed.
5. In Figure 4, the mRNA expressions of the NF-κB downstream target genes including A1a, A20 and c-myc are measured. However, one cannot clearly related those genes to H. pylori-induced inflammation. Are they the pro-inflammatory genes induced by NF-κB? (see the drawings in Figure 5) If so, authors should provide the reasonings and references in the manuscript.
6. The quantitative data of Fig. 4A show that the pIκBα/IκBα in KP+HP group is lower than that of the control. However, it does not match with your western blot!
7. For Fig 4A, It seems that the amount of IκBα is lower in H. pylori group. The quantitative data of IκBα and another housekeeping gene as a loading control should be added for the western blot data to support the schematic representation of anti-inflammatory effects of KP on H. pylori-infected in mice via inhibition of the NF-κB signaling pathway.
8. For all the quantitative data, the number of independent experiments should be provided in the legends.
9. Some of the descriptions and labels for the axis in the compound figure are very small, please enlarge them.
10. There should be hyphens between “KP” and “administrated mice”, “KP” and “treated mice”, and “calcium” and “dependent”. Please check and edit the whole manuscript.
11. Too many capital letters in the legend 1E
12. Check the usage of capital letters in all the subheadings. They need to be consistent.
13. Line 231, check the size of the letter. It need to be consistent throughout the manuscript.
14. Line 313, remove the from “Attachment of the H. pylori”
15. The latter half of the sentence is not clear described. “KP prevented gastric damages by inhibiting H.pylori growth and attenuating virulence factors such as CagA, ureA, SSA and napA”
16. Several sentences are not clear described, especially those in the result section and figure legends. Here, only several sentences are listed below. Please check and edit the whole manuscript.
“Exposure of H. pylori group was significantly increased the inflammation whereas the treatment of KP group was dramatically decreased the damage in gastric tissues (Figure 1E).”
“As shown in Figure 1F, the H. pylori group indicated significantly loss of PAS positive mucus cells in gastric tissue.”
“treatment with KP group showed”
“H. pylori –infected group was significantly increased in H. pylori virulence factor levels”
“KP-treated group was extremely reduced the virulence factor levels”
“we examined ELISA and qPCR analysis”
Author Response
Reviewer 1:
This manuscript describes the potential of using Korean propolis (KP) as an alternative therapeutic approach for the treatment of H. pylori infection. The authors aim to show that KP is able to inhibit the inflammation via suppression of NF-κB signaling on H. pylori-infected mice model. While some findings seem to be interesting, some puzzling features need to be further addressed.
Major comments:
- 1. As indicated in Fig.1A, KP was administrated after the mice was infected with pylori. Why the author stated that “these findings indicated that KP has “preventive effects” for inflammation via suppression of NF-κB signaling on H. pylori-infected mice model.” in the abstract? I do not agree that there is a prevented effect according to the design of your experiments.
→ Author reply: Thank you for your important comments. In accordance with the reviewer’s comment, we checked the words, and we have modified ‘preventive effects’ to ‘therapeutic effects’ on line 21 in the abstract part of revised manuscript. Revised parts are highlighted in manuscript.
- Need to provide the full name for “CLO test”
→ Author reply: In accordance with the reviewer’s comment, we have provided the full name for ‘Campylobacter-Like Organism (CLO) Test’ on line 107 in the Material and Methods part of revised manuscript. Revised parts are highlighted in manuscript.
- The legend of Fig 1C is written as “Results of the CLO test with gastric mucosal following each group”. Since the resolution of Fig 1C is very poor, one cannot see the result at all even using zoon in.
→ Author reply: In accordance with the reviewer’s comment, we have modified the Figure legend 1C in revised manuscript. Also, we have increased the resolution and the size of Fig.1C in the revised manuscript. Revised parts are highlighted in manuscript.
- In Figure 2, authors show that treatment of KP reduces the expression of H. pylori virulence factors in H. pylori-infected gastric mucosal injury mice model. Is the reduction due to the inhibition of the expression of these virulence factors in H. pylorior the eradication of H. pyloriin the mice? This question needs to be addressed.
→ Author reply: Thank you for your important comments. The effect of Korean propolis (KP) against the eradication of H. pylori has already been demonstrated and published (Ref. Song MY et al., J. Microbiol., 2020; Han et al., Korean J. Food Nutr., 2016). However, the effect of KP on the reduction of H. pylori virulence factors has not yet been elucidated. In this study, we showed the results of the changes of H. pylori after Korean propolis (KP) intake using H. pylori IgG test, Campylobacter-like organism (CLO) test and the expression of H. pylori virulence factors. As shown in Figure 1B, the level of H. pylori IgG was significantly increased in mice infected with H. pylori, whereas it was significantly suppressed in KP-treated mice after H. pylori infection. These data suggest that the amount of H. pylori was decreased after KP intake in H. pylori-infected mice. Moreover, as a result of the CLO test used clinically for the cure rate and investigation of H. pylori colonization, as shown in Figure 1C, H. pylori-infected group produced a positive reaction (purple color), however KP-administrated group showed a weak positive or negative reaction (yellow color). These data indicate that KP significantly inhibited the growth and survival of H. pylori, supporting that the intake of KP decreased the amount of H. pylori.
As another indicator, we showed the expressions of mRNA and protein of H. pylori virulence factors using qPCR and Western blot analysis. Lamb et al indicated the H. pylori virulence factors contribute to the flagella, chemotactic systems, ureases, and adhesions upon H. pylori growth (Lamb et al., J. Cell. Biochem., 2013). Chang et al also suggest that virulence factors of H. pylori are associated with colony formation and pathogenic responses (Chang et al., J. Biomed. Sci., 2018). As shown in Figure 2, the expression of H. pylori virulence factor such as 16s rRNA, Sydney strain 1 (SS1), encoding urease A subunit (ureA), surface antigen gene (SSA) and neutrophil-activating protein A (napA) were significantly increased in the H. pylori-infected group. However, the mRNA expression of virulence factors and the protein expression of cagA was dramatically suppressed in KP-treated mice after H. pylori infection compared to the H. pylori-infected mice. These data suggest that KP intake inhibits the virulence factors required for the growth and survival of H. pylori. Therefore, we speculate that the reduction of H. pylori virulence factors is due to the inhibition of H. pylori growth by KP intake. However, since KP cannot completely eradicate H. pylori, our findings suggest that KP exerts anti-inflammatory effect through inhibition of NF-κB signaling in the host on the inflammatory response induced by the remaining H. pylori.
- In Figure 4, the mRNA expressions of the NF-κB downstream target genes including A1a, A20 and c-myc are measured. However, one cannot clearly related those genes to H. pylori-induced inflammation. Are they the pro-inflammatory genes induced by NF-κB? (see the drawings in Figure 5) If so, authors should provide the reasonings and references in the manuscript.
→ Author reply: A1a, A20 and c-myc are representative pro-inflammatory genes known to be regulated by NF-κB signaling pathway in various inflammatory diseases including cancers (Santoro et al., EMBO J., 2003; Moser et al., Mol. Cancer, 2021) In accordance with the reviewer’s comment, we have added these references on line 296-297 in Result section of the revised manuscript. Revised parts are highlighted in manuscript.
- The quantitative data of Fig. 4A show that the pIκBα/IκBα in KP+HP group is lower than that of the control. However, it does not match with your western blot!
→ Author reply: Thank you so much for your comments. In Figure 4A, each sample was measured independent experiments for 4 times and the average of these measurements was taken as quantitative data. The whole data images for the protein expression of p-IκBα are shown below, and we changed the representative Western blot image to match the quantitative data in Figure 4A.
- For Fig 4A, It seems that the amount of IκBα is lower in H. pylori group. The quantitative data of IκBα and another housekeeping gene as a loading control should be added for the western blot data to support the schematic representation of anti-inflammatory effects of KP on H. pylori-infected in mice via inhibition of the NF-κB signaling pathway.
→ Author reply: In accordance with the reviewer’s comment, we have added the quantitative data of IκBα and another housekeeping gene, β-actin as a loading control for the Western blot analysis to support the schematic representation in Figure 4A.
- For all the quantitative data, the number of independent experiments should be provided in the legends.
→ Author reply: In accordance with the reviewer’s comment, we have provided the number of independent experiments in all figure legends. Revised parts are highlighted in manuscript.
- Some of the descriptions and labels for the axis in the compound figure are very small, please enlarge them.
→ Author reply: In accordance with the reviewer’s comment, we checked and enlarged the font size of descriptions and labels for the axis in all figures.
- There should be hyphens between “KP” and “administrated mice”, “KP” and “treated mice”, and “calcium” and “dependent”. Please check and edit the whole manuscript.
→ Author reply: In accordance with the reviewer’s comment, we have revised the whole manuscript and inserted hyphens between ‘KP’ and ‘administrated mice’, ‘KP’ and ‘treated mice’, and ‘calcium’ and ‘dependent.
- Too many capital lettersin the legend 1E
→ Author reply: In accordance with the reviewer’s comment, we have modified the capital letters in the Figure legend 1E. Revised parts are highlighted in manuscript.
- Check the usage of capital letters in all the subheadings. They need to be consistent.
→ Author reply: In accordance with the reviewer’s comment, we have checked the capital letters of all sub-headings in manuscript and corrected consistently.
- Line 231, check the size of the letter. It need to be consistent throughout the manuscript.
→ Author reply: In accordance with the reviewer’s comment, we have checked the font size of all parts and corrected consistently throughout the manuscript.
- Line 313, remove the from “Attachment of the H. pylori”
→ Author reply: We have modified the third paragraph in Discussion part of the revised manuscript. Revised parts are highlighted in manuscript.
- The latter half of the sentence is not clear described. “KP prevented gastric damages by inhibiting H. pylori growth and attenuating virulence factors such as CagA, ureA, SSA and napA”
→ Author reply: In accordance with the reviewer’s comment, we have rewritten the paragraph on line 322-334 and 351-360 in Discussion part of the revised manuscript. Revised parts are highlighted in manuscript.
- Several sentences are not clear described, especially those in the result section and figure legends. Here, only several sentences are listed below. Please check and edit the whole manuscript.
→ Author reply: Thank you so much for your thoughtful comments. In accordance with the reviewer’s comment, we have checked the whole manuscript and revised the sentences in the Result and Figure legends more clearly. Revised parts are highlighted in manuscript.
“Exposure of H. pylori group was significantly increased the inflammation whereas the treatment of KP group was dramatically decreased the damage in gastric tissues (Figure 1E).”
→ We have edited the sentence on line 210-213 in the Result part of revised manuscript. Revised parts are highlighted in manuscript.
“As shown in Figure 1F, the H. pylori group indicated significantly loss of PAS positive mucus cells in gastric tissue.”
→ We have modified the sentence on line 218-221 in the Result part of revised manuscript. Revised parts are highlighted in manuscript.
“treatment with KP group showed”
→ We have amended the sentence on line 218-221 in the Result part of revised manuscript. Revised parts are highlighted in manuscript.
“H. pylori –infected group was significantly increased in H. pylori virulence factor levels”
→ We have improved the sentence on line 236-241 in the Result part of revised manuscript. Revised parts are highlighted in manuscript.
“KP-treated group was extremely reduced the virulence factor levels”
→ We have corrected the sentence on line 241-244 in the Result part of revised manuscript. Revised parts are highlighted in manuscript.
“we examined ELISA and qPCR analysis”
→ We have rewritten the sentence on line 261-263 in the Result part of revised manuscript. Revised parts are highlighted in manuscript.

Reviewer 2 Report
The topic of the manuscript entitled"Anti-inflammatory effect of Korean propolis on Helicobacter pylori-infected gastric mucosal injury mice model" is interesting, and the manuscript can give us a further unstanding of the Korean propolis. The manuscript is well designed and presented.
1. Check the reference format carefully;
2. The words in the figures are not clear;
3. Authors should add n=? after M±SD in the figure legand;
4. Other revisions are marked in the text.

Author Response
Reviewer 2:
The topic of the manuscript entitled "Anti-inflammatory effect of Korean propolis on Helicobacter pylori-infected gastric mucosal injury mice model" is interesting, and the manuscript can give us a further unstanding of the Korean propolis. The manuscript is well designed and presented.
→ Author reply: Thank you so much for your favorable comments.
Major comments:
- Check the reference format carefully;
→ Author reply: In accordance with the reviewer’s comment, we have checked all reference format of the manuscript and edited correctly for the journal style.
- The words in the figures are not clear;
→ Author reply: In accordance with the reviewer’s comment, we have modified the size of words in all figures more clearly.
- Authors should add n=? after M±SD in the figure legand;
→ Author reply: In accordance with the reviewer’s comment, we have added the number of animals or samples in the experiment to all Figure legends. Revised parts are highlighted in manuscript.
- Other revisions are marked in the text.
→ Author reply: In accordance with the reviewer’s revisions, we have checked and modified the marked in the text. Revised parts are highlighted in Abstract, Material and Methods, Result, Table 1, Reference parts of the revised manuscript.

Reviewer 3 Report
The effect of propolis on HP (Helicobacter pylori) is a critical issue in food and nutrition, but the results mainly focus on gastric mucosal injury, not HP.
The major problem: HP changes after propolis intake need to be provided, but the main results are anti-inflammatory effects.
Another problem: the effect of propolis on HP is a common reports, so author need to add their highlights.
Author Response
Reviewer 3:
The effect of propolis on HP (Helicobacter pylori) is a critical issue in food and nutrition, but the results mainly focus on gastric mucosal injury, not HP.
Major comments:
- The major problem: HP changes after propolis intake need to be provided, but the main results are anti-inflammatory effects.
→ Author reply: Thank you for your important comments. In this study, we showed the results of the changes of H. pylori after Korean propolis (KP) intake using H. pylori IgG test, Campylobacter-like organism (CLO) test and the expression of H. pylori virulence factors.
The level of H. pylori IgG is a classic indicator of H. pylori infection. Li et al demonstrated that anti-H. pylori IgG responses showed the value of clinical presentations in diagnosis of H. pylori infection (Li et al., World J. Gastroenterol., 2003). When H. pylori is infected, the IgG levels are significantly higher than in the non-infected group. As shown in Figure 1B, the level of H. pylori IgG was significantly increased in mice infected with H. pylori, whereas it was significantly suppressed in KP-treated mice after H. pylori infection. These data suggest that the amount of H. pylori was decreased after KP intake in H. pylori-infected mice.
The CLO test is also known as a classic indicator of H. pylori infection. Unver et al reported that the CLO test was sensitive and rapidly detected for H. pylori (Unver et al., Laryngoscope, 2001). Moreover, it has been clinically used for the cure rate and investigation of H. pylori colonization. As shown in Figure 1C, H. pylori-infected group produced a positive reaction (purple color), however KP-administrated group showed a weak positive or negative reaction (yellow color). These data indicate that KP significantly inhibited the growth and survival of H. pylori, supporting that the intake of KP decreased the amount of H. pylori.
As another indicator, we showed the expressions of mRNA and protein of H. pylori virulence factors using qPCR and Western blot analysis. Lamb et al indicated the H. pylori virulence factors contribute to the flagella, chemotactic systems, ureases, and adhesions upon H. pylori growth (Lamb et al., J. Cell. Biochem., 2013). Chang et al also suggest that virulence factors of H. pylori are associated with colony formation and pathogenic responses (Chang et al., J. Biomed. Sci., 2018). As shown in Figure 2, the expression of H. pylori virulence factor such as 16s rRNA, Sydney strain 1 (SS1), encoding urease A subunit (ureA), surface antigen gene (SSA) and neutrophil-activating protein A (napA) were significantly increased in the H. pylori-infected group. However, the mRNA expression of virulence factors and the protein expression of cagA was dramatically suppressed in KP-treated mice after H. pylori infection compared to the H. pylori-infected mice. These data suggest that KP intake inhibits the virulence factors required for the growth and survival of H. pylori.
- Another problem: the effect of propolis on HP is a common reports, so author need to add their highlights.
→ Author reply: In the present study, we focused on the effects of Korean propolis (KP) in a mouse model infected with H. pylori. The effect of propolis on H. pylori has already been reported (Baltas et al., J. Enzyme Inhib. Med. Chem., 2016; Shapla et al., J. Appl. Biomed., 2018). However, the reports so far have mainly focused on the efficacy of propolis on H. pylori growth inhibition or eradication. In addition, although there are many reports of propolis in countries such as Brazil, Europe, North America, Cuba, Asia-Pacific region, there are few reports of Korean propolis. Therefore, here, we showed for the first time the efficacy and the mechanism of propolis extracted from South Korea against H. pylori-induced gastric injury.
Kujumgiev et al. reported that propolis has a variety of mixtures for different pharmacological composites depending on the geographic region from which it was extracted (Kujumgiev et al., J. Ethnopharmacol., 1999). Therefore, the potency of compounds contained in propolis would also vary depending on the country of origin. Ahn and Han have already shown that KP has an abundant of total phenolic content and has a strong effect on in vitro experiments (Ahn et al., J. Agric. Food Chem. 2004; Han et al., Korean J. Food & Nutr., 2016). In addition, our previous study revealed the effect of KP on H. pylori in gastric epithelial cells (Song et al., J. Microbiol. 2020), but the effect of KP on H. pylori in experimental animal model has not yet been published. Therefore, we conducted this study using experimental animal model.
Finally, for the treatment of H. pylori-infected diseases, it is important to investigate the efficacy and mechanisms of the host response. Thus, we provided the effects on KP as well as the molecular mechanism of KP on H. pylori-infected mice.

Round 2
Reviewer 3 Report
The present manuscript meets the Nutrients publication requirements. But are the IgG test, CLO test, and HP virulence factors the major HP evaluation? Why not the microbial culture and counting?